# Upregulation of Inflammatory Mediators in Peripheral Blood CD40^+^ Cells in Children with Autism Spectrum Disorder

**DOI:** 10.3390/ijms24087475

**Published:** 2023-04-19

**Authors:** Abdullah A. Aldossari, Mushtaq A. Ansari, Ahmed Nadeem, Sabry M. Attia, Saleh A. Bakheet, Laila Y. Al-Ayadhi, Mohammed M. Alanazi, Mudassar Shahid, Mohammad Y. Alwetaid, Marwa H. Hussein, Sheikh F. Ahmad

**Affiliations:** 1Department of Pharmacology and Toxicology, College of Pharmacy, King Saud University, Riyadh 11451, Saudi Arabia; 2Department of Physiology, College of Medicine, King Saud University, Riyadh 11451, Saudi Arabia; 3Department of Pharmaceutics, College of Pharmacy, King Saud University, Riyadh 11451, Saudi Arabia; 4Department of Botany and Microbiology, College of Science, King Saud University, Riyadh 11451, Saudi Arabia

**Keywords:** autism spectrum disorder, cytokines, CD40, CCR1, PBMCs, transcription factors

## Abstract

Autism spectrum disorder (ASD) is a common and severe neurodevelopmental disorder in early childhood, defined as social and communication deficits and repetitive and stereotypic behaviours. The aetiology is unknown in most cases. However, several studies have identified immune dysregulation as potentially promoting ASD. Among the numerous immunological findings in ASD, reports of increased pro-inflammatory markers remain the most consistently observed. C-C chemokine receptor type 1 (CCR1) activation is pro-inflammatory in several neurological disorders. Previous evidence has implied that the expression of chemokine receptors, inflammatory mediators, and transcription factors play a pivotal role in several neuroinflammatory disorders. There have also been reports on the association between increased levels of proinflammatory cytokines and ASD. In this study, we aimed to investigate the possible involvement of CCR1, inflammatory mediators, and transcription factor expression in CD40^+^ cells in ASD compared to typically developing controls (TDC). Flow cytometry analysis was used to determine the levels of CCR1-, IFN-γ-, T-box transcription factor (T-bet-), IL-17A-, retinoid-related orphan receptor gamma t (RORγt-), IL-22- and TNF-α-expressing CD40 cells in PBMCs in children with ASD and the TDC group. We further examined the mRNA and protein expression levels of CCR1 using real-time PCR and western blot analysis. Our results revealed that children with ASD had significantly increased numbers of CD40^+^CCR1^+^, CD40^+^IFN-γ^+^, CD40^+^T-bet^+^, CD40^+^IL-17A^+^, CD40^+^RORγt^+^, CD4^+^IL-22^+^, and CD40^+^TNF-α^+^ cells compared with the TDC group. Furthermore, children with ASD had higher CCR1 mRNA and protein expression levels than those in the TDC group. These results indicate that CCR1, inflammatory mediators, and transcription factors expressed in CD40 cells play vital roles in disease progression.

## 1. Introduction

Autism spectrum disorder (ASD) is a complex heterogeneous neurodevelopmental disorder characterised by social interaction, communication impairments, and several social and behavioural abnormalities [1]. The aetiological conditions of ASD are still being explored, but previous studies have suggested multifactorial aetiopathogenesis resulting from interactions between genomic, genetic, and environmental factors associated with its development [2,3]. The immune system plays an important role in neurodevelopment, and immune abnormalities have been observed in patients with ASD [4,5]. Cytokine profiles are associated with changes in behavioural symptoms in a subset of individuals with ASD [6]. Previous studies have shown that abnormal cytokine production in individuals diagnosed with ASD is associated with abnormal cytokine levels [7]. Elevated plasma cytokine levels in ASD provide evidence of immune dysfunction and are associated with impaired behavioural outcomes [8]. We previously demonstrated immune aberrations through an imbalance in pro- and anti-inflammatory cytokines in children with ASD [9]. We have further reported the dysregulation of Th1, Th2, Th17, and T regulatory cell-related transcription factor signalling in children with ASD [10]. We also reported that the upregulation of the JAK-STAT signalling pathways in children with ASD could play a critical role in immune dysfunction in ASD [11]. The upregulation of peripheral chemokines and chemokine receptor expression is associated with immune dysregulation in children with ASD [12]. However, the exact underlying mechanism requires further investigation.

C-C chemokine receptor type 1 (CCR1) is a member of the beta chemokine receptor family [13]. Chemokine receptors are associated with several behavioural impairments in individuals with ASD [14]. Several studies have shown that chemokine receptor expression is higher in the brain tissue of patients with ASD [15,16]. Chemokine receptors are expressed on neurons, upregulated in neurological disorders [17], and are functional mediators of neuroinflammatory disorders [18]. Higher levels of chemokine receptors have been found in astrocytes, the anterior cingulate gyrus, the cerebellum, and the brain of patients with autism [19]. The activation of CCR1 is pro-inflammatory in neurological diseases, and it is expressed in various cell types in the brain, including smooth muscle cells, astrocytes, and neurons [20,21]. CCR1 activation contributes to the blood−brain barrier (BBB) damage associated with various neuroinflammatory diseases [22]. CD40 is a TNF receptor (TNFR) superfamily member mainly expressed in APCs such as B cells, macrophages, and dendritic cells [23]. CD40 is presented on the surface of B and T-cells, respectively, which, upon ligation, activates B-cells for antibody isotype switching and upregulates the production of pro-inflammatory cytokines. While most neurological diseases have several factors contributing to pathogenesis, aberrant neuroinflammation mediated by CD40 increases BBB permeability, exacerbates edema and neuronal and glial cell damage, and promotes the formation of occlusive microthrombi [24]. The CD40 signalling pathway has been well documented as an immune checkpoint and humoral and cellular immune stimulator. The CD40 signalling pathway in APCs contributes to numerous cellular functions such as Th17 cell polarisation, proinflammatory cytokine release, and immunoglobulin isotype switching [25]. Several studies have demonstrated the involvement of CD40 cells in the pathogenesis of systemic lupus erythematosus and BTBR mouse model of ASD [26,27]. A recent study showed that CD40 cells significantly increased in the BTBR mouse model of ASD [28]. Therefore, this study explored the overall inflammatory potential of APCs expressing CD40 in children with ASD.

Abnormal IFN-γ production is associated with several immune diseases. Previous studies have shown an increased production of inflammatory mediators in the brains of individuals with ASD [29]. Studies of blood from individuals with ASD showed that IFN-γ was significantly increased compared to controls [30]. A high level of prenatal IFN-γ increases the risk of having an ASD child [31]. Previous results revealed the unexpected role of IFN-γ in regulating neuronal connectivity and social behaviour [32]. T-box transcription factor (T-bet) is essential for differentiating Th1 cells [33]. T-bet expression is important in inflammatory responses and is critical in autoimmune diseases [34]. In humans, T-bet expression is induced in B cells [35], and a recent study has suggested that T-bet-expressing B cells are recruited to the CNS [36]. T-bet increases IL-17A levels in the CNS and T-cell infiltration, concomitant with neuroinflammation [37].

IL-17A is a key regulator of ASD development [38], and elevated IL-17 and IL-17A have been observed in children with ASD [39,40]. It has also been reported that autism-like symptoms in the offspring of immune-activated maternal mice may be due to IL-17A signalling [41]. The enrichment of IL-17 genes has been reported in individuals with ASD, indicating a possible role for this cytokine in the pathophysiology of this condition [42]. Retinoid-related orphan receptor gamma t (RORγt), a master transcription factor for Th17, is critical in mediating many autoimmune diseases [43]. Interleukin (IL)-22 is a potent mediator of inflammatory responses. Th22 cells are a major source of IL-22 and play an important role in several neurological disorders by promoting leukocyte infiltration in the brain [44]. IL-22 also regulates chemo-attractant production by microvascular endothelial cells in the BBB [45], plays a role in several human diseases, and its upregulation is associated with lymphocyte activation in the CNS [46]. TNF-α is a central regulator of inflammation and is elevated in the cerebrospinal fluid of children with ASD [47]. In addition, TNF-α expression is elevated in ASD, suggesting a dysregulated immune response [48]. Previous studies have shown that a high level of TNF-α is correlated with the severity of ASD symptoms [8,15,49]. These results indicate the potential role of inflammatory mediators in neuroimmune dysfunction. We hypothesised that CD40 expression promotes an immune imbalance in children with ASD. Thus, the restoration of CD40 may be considered a treatment strategy for immune abnormalities in children with ASD.

## 2. Results

### 2.1. Upregulation of CCR1-Expressing CD40^+^ Cells in Children with ASD

Flow cytometry was performed to evaluate the number of CCR1-expressing CD40^+^ cells in the ASD and TDC groups. The number of CCR1-expressing CD40^+^ PBMCs was elevated in the children with ASD compared to that in the TDC group (Figure 1A). We also evaluated the gene expression of CCR1 from the ASD and TDC groups. The children with ASD showed a significant increase in CCR1 mRNA expression compared to the TDC group (Figure 1B). We further evaluated CCR1 protein expression levels in PBMCs. The protein expression level of CCR1 was upregulated in the children with ASD compared to the TDC group (Figure 1C). Our results suggest an association between chemokine receptors and ASD development. 

### 2.2. Elevation of IFN-γ- and T-Bet-Expressing CD40^+^ Cells in Children with ASD

We evaluated the number of CCR1-expressing CD40^+^ cells in PBMCs from the children with ASD and the TDC group. Our results showed that the number of IFN-γ-expressing CD40^+^ cells was significantly higher in the children with ASD than in the TDC group (Figure 2A). We further evaluated the number of T-bet-expressing CD40^+^ cells in the PBMCs. We observed increased numbers of T-bet-expressing CD40^+^ cells in the children with ASD compared with those in the TDC group (Figure 2B). These results show that increased levels of inflammatory cytokines and transcription factors could be linked to the severity of immune dysfunction in children with ASD.

### 2.3. Upregulation of IL-17A- and RORγt-Expressing CD40^+^ Cells in Children with ASD

This study aimed to understand the contributions of cytokines and transcription factors in children with ASD. Our results showed that several IL-17A-expressing CD40^+^ cells were significantly upregulated in the children with ASD compared with those in the TDC group (Figure 3A). Additionally, our results showed that the number of RORγt-expressing CD40^+^ cells was higher in the children with ASD as compared with TDC group (Figure 3B). Based on these results, ASD development may be associated with alterations in cytokine and transcription factor signalling.

### 2.4. Upregulation of IL-22- and TNF-α-Expressing CD40^+^ Cells in Children with ASD

Our results further defined the number of IL-22- and TNF-α-expressing CD40^+^ cells in PBMCs from the children with ASD and the TDC group. We demonstrated that the number of IL-22-expressing CD40^+^ cells was higher in the children with ASD than in the TDC group (Figure 4A). Furthermore, our results showed that the children with ASD had significantly increased TNF-α-expressing CD40^+^ cells as compared to the TDC group (Figure 4B). Our results provide evidence for the upregulation of IL-22- and TNF-α expression in CD40 cells in children with ASD. According to the Pearson correlation coefficient, there was no correlation between the severity of symptoms and different immune parameters.

## 3. Discussion

Emerging evidence suggests that altered communication between the nervous system and inflammatory pathways is associated with multiple diseases, including ASD. The nervous and immune systems constantly communicate [50]. Multiple lines of evidence have recently pinpointed the key contribution of B lymphocytes to ASD pathogenesis. Altered immune responses commonly occur in individuals genetically susceptible to ASD [51,52]. Several studies have reported the involvement of immune dysregulation in the pathophysiology of ASD [53]. It has also been suggested that immune alterations contribute to behavioural effects in neurodevelopmental disorders, including ASD [54,55]. Evidence shows that dysregulation of the immune balance is a high risk factor for neurodevelopmental defects in ASD [56]. Several dysregulated cytokines and transcription factors in ASD have also been correlated with symptom severity and performance in ASD diagnostic tests [1,9]. Several chemokines recruit other immune cells to sites of tissue damage or infection. A previous study indicated an association between impaired behaviour and elevated chemokine levels in ASD [15]. Thus, cytokine, chemokine receptor, and transcription factor dysregulation could have important biological effects on neuronal development and activity that adversely affect behaviour.

It is well known that chemokines and their receptors play an important role in the immune system. Accumulating evidence has revealed that chemokine receptor expression, distribution, and function are involved in the pathogenesis of neurodegenerative diseases [57,58]. Chemokine receptors have been identified as regulators of peripheral immune cell trafficking and are expressed in the CNS [59,60]. Several studies have reported the expression of CCR1 in neurons, microglia, and astrocytes [61,62]. CCR1 mRNA and protein levels are increased in the brain, spinal cord, peripheral lymphoid organs, and blood plasma [63]. CCR1 promotes the entry of immune cells into the brain during neuroinflammation [64]. In the present study, we analysed CCR1 expression, an important proinflammatory chemokine among the CC chemokines, in children with ASD. Our results showed that CCR1-expressing CD40^+^ cells were significantly increased in the children with ASD compared with those in the TDC control group. Moreover, the children with ASD had significantly elevated CCR1 mRNA and protein expression compared to the TDC controls, indicating elevated levels of CCR1 expression. Our study provides strong evidence of the role of chemokine receptors in children with ASD. Therefore, chemokine receptors may be clinically useful disease markers for ASD. These observations may be highly relevant for children with ASD and other neuroimmune disorders.

A recent study described that the IFN-γ level was increased in the ASD brain [65]. It has been shown that IFN-γ level is elevated in children with ASD [9,28]. Consistent with the role of IFN-γ expression in ASD, the children with ASD had significantly increased IFN-γ expression levels [66]. Previous studies showed that prenatal IFN-γ imbalances could be linked to autism [31]. Early reports suggested a strong association with high levels of IFN-γ in patients diagnosed with ASD [67,68], which have been implicated in the pathophysiology of these neurobehavioural diseases [69]. Higher levels of cytokines, including IFN-γ, are demonstrated in ASD [8,70]. T-bet regulates Th1 and Th17 lymphocytes and infiltrates the T cells associated with CNS neuroinflammation [71,72]. T-bet plays an important role in disease development and is expressed in the CNS-infiltrating T cells [73,74]. T-bet-expressing cells are encephalitogenic in the CNS, and their infiltration is associated with neuroinflammation [37,75]. Thus, cytokine dysregulation could have important biological effects on neuronal development, adversely affecting behaviour. Our study showed that IFN-γ- and T-bet-expressing CD40^+^ cells were increased in children with ASD. In our study, CD40^+^IFN-γ^+^ and CD40^+^T-bet^+^ cells appeared more highly represented in aggressive ASD children. The exact mechanism by which IFN-γ- and T-bet-expressing CD40^+^ cells are involved in neuroinflammation remains to be explored. Further studies are needed to link these factors with disease severity. Therefore, increased cytokine and transcription factor levels may be associated with impaired communication and aberrant behaviour. These observations are highly relevant to children with ASD and other neuroimmune disorders. Therefore, further studies are needed to elucidate the association between proinflammatory cytokines and transcription factors in patients with ASD.

A recent study showed that IL-17A expression is significantly upregulated in the peripheral immune cells of children with ASD [9]. IL-17A is associated with behavioural impairments in ASD, suggesting that peripheral inflammation influences neuronal development [8,76]. Previous studies have shown that increased levels of IL-17A are associated with the severity of ASD behavioural symptoms [38,54]. In murine models, it has been shown that IL-17A plays a significant role in the induction of autism-like symptoms in the offspring of immune-activated mothers [16,77]. These parallel malformations are abnormalities found in the brain development of children with ASD [78,79]. RORγt, the key Th17 cell transcriptional regulator, is associated with neurodegeneration [80]. It has also been shown that RORγt expression is significantly upregulated in children with ASD and BTBR mice [9,40]. RORγt expression also correlates with IL-17 production, and the impact of suppressing RORγt could serve as a more effective treatment for neuroinflammation [81]. Recent findings suggest that pathogenic CD4^+^RORγt^+^ cells contribute to brain inflammation and neurobehavioural disorders [82]. Therefore, IL-17A dysregulation may play a central role in the development of ASD. Our results indicate that IL-17A- and RORγt-expressing CD40^+^ are upregulated in the children with ASD compared to the TDC control group. Therefore, we hypothesised that proinflammatory mediators and their transcription factors are involved in behavioural aggravation and neuroimmune dysfunction in children with ASD. These results provide evidence that IL-17A/RORγt expression in CD40 cells could be associated with immune and neuronal dysfunction in ASD, and further study is warranted. These observations suggest that dysfunctional immune responses may affect the core features of ASD and its associated behaviours in children.

IL-22 is a potent proinflammatory cytokine. IL-22 plays a critical role in human diseases, and its overexpression is associated with lymphocyte activation in the brain [83]. Previous studies have suggested that IL-22 overexpression is associated with immune dysfunction in children with ASD [84]. Another study has suggested that increased IL-22 expression promotes leukocyte infiltration into the brain [85], and increased cytokine IL-22 levels have been confirmed to elevate neurodegenerative disorders [86]. A more recent report noted that an increase in TNF-α level is a potentially important biomarker in ASD [87,88]. Previous studies have also demonstrated that TNF-α increases in various tissues of patients with ASD [89], and TNF-α expression is elevated in children with ASD [90]. It is also interesting to note that several studies also demonstrated increased TNF-α production in children with ASD [91,92]. Importantly, TNF-α crosses from the peripheral blood into the brain, directly affecting brain function and behaviour [93,94]. To our knowledge, there have been no investigations of IL-22- and TNF-α-expression CD40^+^ cells in ASD; therefore, our current study examined IL-22/TNF-α expression of CD40^+^+ cells in children with ASD. Recent results have also shown that the number of IL-22-expressing CD40^+^ PBMCs is significantly higher in children with ASD [95]. In the present study, we observed that the children with ASD exhibited increased IL-22 and TNF-α expression in CD40^+^ cells. Therefore, dysfunctional immune defences and inflammatory reactions in children with ASD may be important precipitating factors that trigger this disorder. We speculate that IL-22/TNF-α expression could be used as a clinical marker for ASD, although this requires further study. Our findings suggest that these factors may serve as diagnostic markers and therapeutic targets for diagnosing and treating ASD.

## 4. Materials and Methods

### 4.1. Study Participants

All of the children were assessed by trained ASD clinicians. A total of 30 male children were diagnosed with ASD (mean ± SD = 6.5 ± 2.8 years) based on the Diagnostic and Statistical Manual of Mental Disorders, 5th edition [96]. Children with ASD were enrolled at the Autism Research Treatment Center of King Saud University, Saudi Arabia. The patients included in this study had no associated neurological disorders (such as seizures and tuberous sclerosis) or metabolic disorders (such as phenylketonuria), and patients with known medical conditions were excluded from the study because these comorbidities associated with ASD may have influenced the results. Additionally, the included patients were not receiving any medications.

A total of 24 healthy male children (mean ± SD = 6.1 ± 3.1 years) were enrolled as typically developing controls (TDCs), who attended a routine follow-up at the Well Baby Clinic, College of Medicine, King Khalid Hospital. The TDC children demonstrated no clinical findings suggestive of neuropsychiatric disorders. None of the participants had a history of a recent infection or fever. All of the processes complied with the National Institutes of Health guidelines and the legal requirements of King Saud University for studies involving human subjects. Additionally, the participants’ parents or legal guardians signed an informed written consent form in order for participation.

### 4.2. Study Measurements

Clinical evaluations of the children with ASD were performed based on their history, as obtained from clinical and neuropsychiatric examinations. The severity of the disease was assessed using the Childhood Autism Rating Scale [97], which rates children on a scale from one to four in each of 16 areas: relating to people, listening response, verbal and non-verbal communication, emotional and visual responses, consistency of intellectual response, fear or nervousness, touch and smell responses, imitation, adaptation to change, body use, object use, taste, activity level, and general impressions.

### 4.3. Chemicals and Antibodies

Fluorescein isothiocyanate, phycoerythrin; allophycocyanin; allophycocyanin-Cy7 and PE/Dazzle labelled CD40, CCR1, IFN-γ, T-bet, IL-17A, IL-22, and TNF-α anti-human monoclonal antibodies; and red blood cell lysing, fixation, and permeabilizing buffers were purchased from BioLegend (San Diego, CA, USA). FcR blocking reagent was purchased from Miltenyi Biotech (Bergisch Gladbach, Germany). RORγt, GolgiStop, and acid−citrate−dextrose vacutainer tubes were purchased from BD Biosciences (San Diego, CA, USA). The primers used were purchased from GenScript (Piscataway, NJ, USA). TRIzol reagent was obtained from Life Technologies (Grand Island, NY, USA). The SYBR Green PCR Master Mix and High-Capacity cDNA Reverse Transcription kit was purchased from Applied Biosystems (Paisley, UK). The primary antibodies against CCR1 and secondary anti-human antibodies used for Western blotting were purchased from Santa Cruz Biotechnology, Inc. (Dallas, TX, USA). Phorbol 12-myristate 13-acetate (PMA), ionomycin, phosphate-buffered saline (PBS), RPMI-1640 medium, Ficoll-Paque, and Hanks’ Balanced Salt Solution (HBSS) were purchased from Sigma-Aldrich (St. Louis, MO, USA). Nitrocellulose membranes were obtained from Bio-Rad Laboratories (Hercules, CA, USA), and a chemiluminescence Western blot detection kit was purchased from GE Healthcare Life Sciences (Piscataway, NJ, USA).

### 4.4. Preparation of PBMCs

Peripheral blood was obtained from the children with ASD and the TDC children in acid−citrate−dextrose vacutainer tubes (BD Biosciences, San Jose, CA, USA). As described previously, PBMCs were isolated using the Ficoll-Paque (specific gravity 1.077; Sigma-Aldrich, St. Louis, MO, USA) gradient density method [9,98]. The peripheral blood samples were centrifuged at 850× *g* for 10 min to remove the plasma. Blood cells were diluted with PBS and centrifuged in a Ficoll-Paque discontinuous gradient at 420× *g* for 30 min. The PBMC layer was collected and washed with cold distilled water and 10 × HBSS to remove red blood cells. As previously described, the cells were resuspended at 2 × 10^6^ cells/mL in RPMI-1640 medium [9,26].

### 4.5. Flow Cytometric Analysis

Flow cytometric analysis was used to measure the number of CCR1-, IFN-γ-, T-bet-, IL-17A-, RORγt, IL-22-, and TNF-α-expressing CD40^+^ cells in the PMBCs. Briefly, PBMCs were stimulated with PMA/ionomycin (Sigma-Aldrich, St. Louis, MO, USA) in the presence of a Golgi stop (BD Bioscience, San Jose, USA), as previously described [12,98]. PBMCs were washed and surface-stained with anti-CCR1 and anti-CD40 antibodies (BioLegend, San Diego, CA, USA). For the staining of cytokines and transcription factors, the cells were fixed, permeabilised, and stained for anti-IFN-γ, anti-T-bet, anti-IL-17A, anti-RORγt, anti-IL-22, and anti-TNF-α (BioLegend, San Diego, CA, USA). Human lymphocytes were isolated from the other immune cells (monocytes and granulocytes) using a conventional gating strategy based on their physical properties (forward and side scatter) to determine different immune markers in the lymphocytes. Chemokine receptors, inflammatory mediators, and transcription factors were identified based on the immunofluorescence characteristics of the antibody-labelled cells in the lymphocyte gate. The proportions of CD40^+^CCR1^+^, CD40^+^IFN-γ^+^, CD40^+^T-bet^+^, CD40^+^IL-17A^+^, CD40^+^RORγt^+^, CD40^+^IL-22^+^, and CD40^+^TNF-α^+^, cells were determined in the lymphocyte gate. Flow cytometry was conducted using an FC500 flow cytometer with CXP software (Beckman Coulter, Brea, CA, USA).

### 4.6. Gene Expression

Real time-PCR (RT-PCR) was performed as described previously [99]. The total RNA was extracted from PBMCs using TRIzol reagent (Invitrogen, Life Technologies, Waltham, MA, USA) and quantified as previously described [12,98]. cDNA synthesis was performed using a high-capacity cDNA reverse transcription kit (Applied Biosystems, Waltham, MA, USA), as previously described [11]. Quantitative analysis of the mRNA expression was performed using RT-PCR. The primers used in this study were purchased from GenScript (Piscataway, NJ, USA). CCR1 Forward: 5’-AATGTAATGGTGGCCTGGGG-3’; Reverse: 5′-TCCTCCCAACCCCCTATCAG-3’; GAPDH Forward: 5’-CTTTGCAGCAATGCCTCCTG-3’; GAPDH Reverse: 5’-ACCATGAGTCCTTCCACGAT-3’. Relative changes in the gene expression were determined using the 2^−ΔΔCT^ method [100] with GAPDH as the reference gene.

### 4.7. Western Blotting

The total cellular protein was extracted from PBMCs using a previously described method [12,26]. Briefly, 40 µg of the isolated protein from PBMC was separated using 7% SDS-PAGE, followed by transfer to the nitrocellulose membrane (Bio-Rad, Hercules, CA, USA). The membrane was blocked in a blocking solution overnight at 4 °C, followed by incubation with a primary CCR1 antibody and secondary peroxidase-conjugated antibody (Santa Cruz Biotech, Dallas, TX, USA) at room temperature for 2 h. The CCR1 and β-actin bands were visualised using a Western blotting luminol reagent (Santa Cruz Biotechnology, Inc., Dallas, TX, USA) and quantified relative to the β-actin bands. Images were obtained using a ChemiDoc Imaging System (Bio-Rad, Hercules, CA, USA).

### 4.8. Statistical Analysis

The Student’s *t*-test was used to compare the two groups. Statistical analyses were performed using GraphPad Prism. The data were expressed as mean ± SD. Statistical significance was set at *p* < 0.05. Pearson’s correlation coefficient ‘r’ was used to assess the relationships between different immune parameters and CARS scores in ASD patients.

## 5. Conclusions

This is the first study demonstrating that CD40 expresses CCR1, proinflammatory cytokines, and other transcription factors in children with ASD. These findings suggest that CCR1, proinflammatory cytokines, and transcription factors may be associated with behavioural disturbances and disease severity in children with ASD. Therefore, the description of immunological parameters in ASD has important implications for diagnosis and should be considered when designing therapeutic strategies to treat the core symptoms and behavioural impairments in ASD.

## Figures and Tables

**Figure 1 ijms-24-07475-f001:**
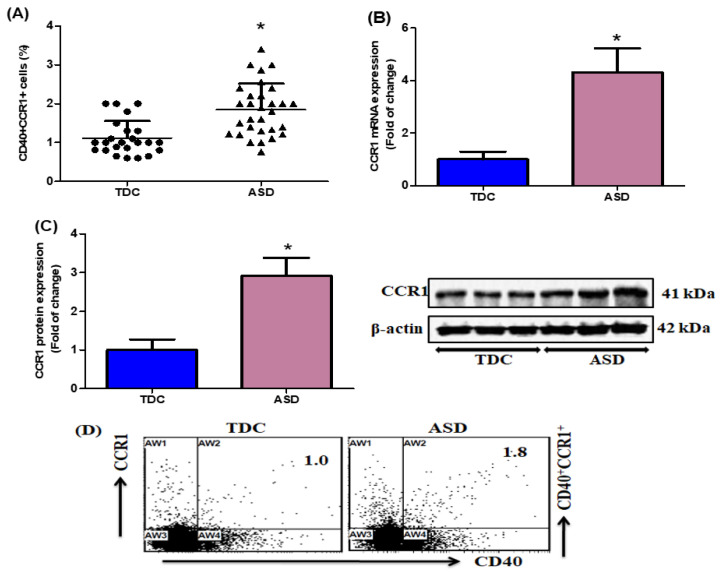
(**A**) Flow cytometric analysis was used to determine the number of C-C chemokine receptor type 1 (CCR1)-expressing CD40^+^ cells in the PBMCs of children with autism spectrum disorder (ASD) and the typically developing controls (TDC) group. (**B**) The mRNA expression level of CCR1 in PBMCs was measured using quantitative RT–PCR and normalized to GAPDH. (**C**) Western blot analysis of CCR1 protein expression in the PBMCs. (**D**) Representative flow cytometry dot plots demonstrating CD40^+^CCR1^+^ cells from the ASD and TDC groups. Statistically significant differences (* *p* < 0.05) were tested using the Student’s *t*-test.

**Figure 2 ijms-24-07475-f002:**
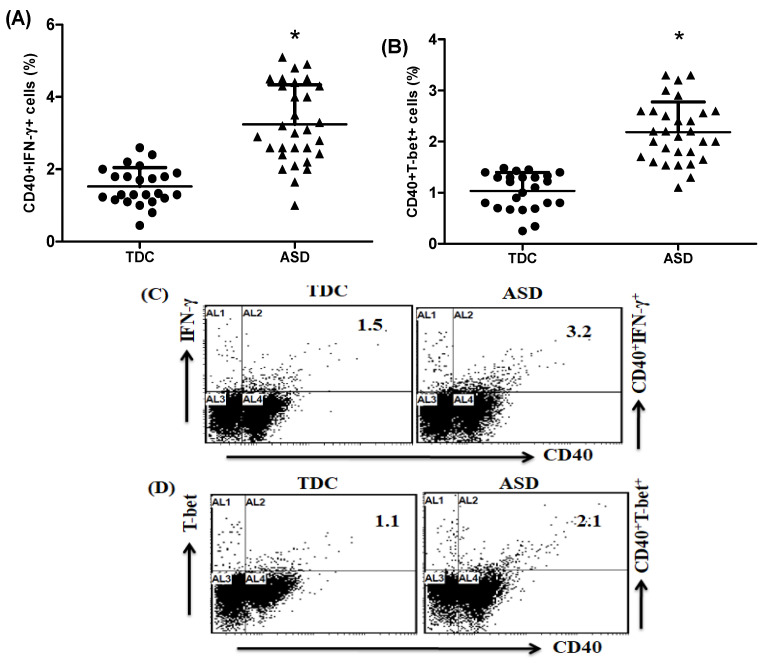
(**A**,**B**) Flow cytometric analysis was used to determine the number of IFN-γ- and T-box transcription factor (T-bet)-expressing CD40^+^ cells in the PBMCs of children with autism spectrum disorder (ASD) and the typically developing controls (TDC) group. (**C**,**D**) Representative flow cytometry dot plots demonstrating CD40^+^IFN-γ^+^ and CD40^+^T-bet^+^ cells from the ASD and TDC groups. Statistically significant differences (* *p* < 0.05) were tested using Student’s *t*-test.

**Figure 3 ijms-24-07475-f003:**
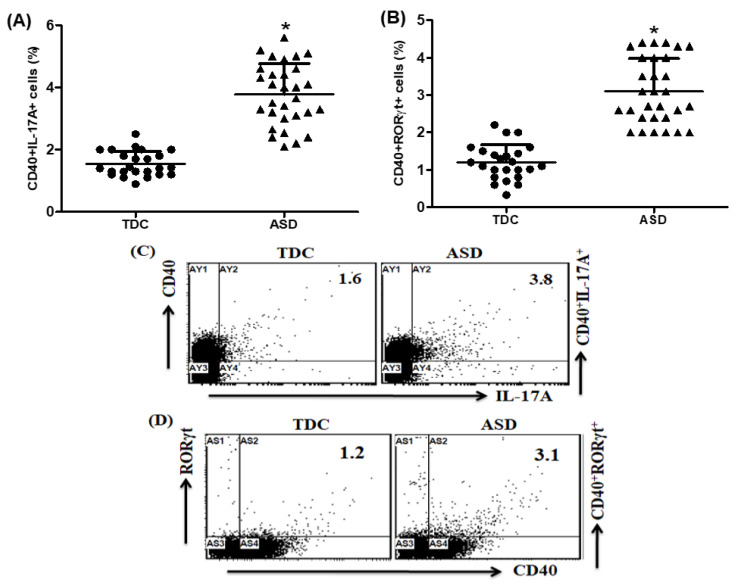
(**A**,**B**) Flow cytometric analysis were used to determine the number of IL-17A- and retinoid-related orphan receptor gamma t (RORγt)-expressing CD40^+^ cells in the PBMCs of children with autism spectrum disorder (ASD) and the typically developing controls (TDC) group. (**C**,**D**) Representative flow cytometry dot plots demonstrating IL-17A- and RORγt-expressing CD40^+^ cells from the ASD and TDC groups. Statistically significant differences (* *p* < 0.05) were tested using the Student’s *t*-test.

**Figure 4 ijms-24-07475-f004:**
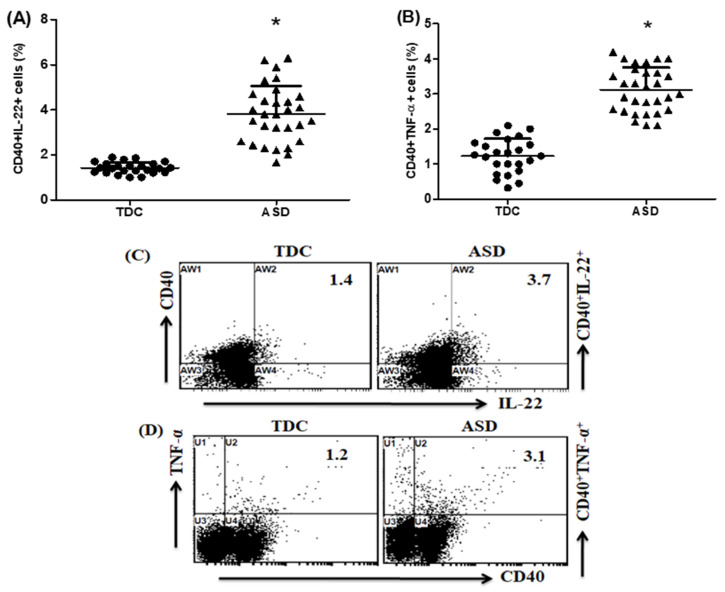
(**A**,**B**) Flow cytometric analysis determined the number of IL-22- and TNF-α-expressing CD40^+^ cells in PBMCs of children with autism spectrum disorder (ASD) and the typically developing controls (TDC) group. (**C**,**D**) Representative flow cytometry dot plots demonstrating IL-22- and TNF-α-expressing CD40+ cells from the ASD and TDC groups. Statistically significant differences (* *p* < 0.05) were tested using the Student’s *t*-test.

## Data Availability

All data presented in this study are available upon reasonable request from the corresponding author.

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
