# Peer review of "Upregulation of Inflammatory Mediators in Peripheral Blood CD40+ Cells in Children with Autism Spectrum Disorder"

_ijms, 2023, doi:10.3390/ijms24087475_

Round 1
Reviewer 1 Report
Aldossari et al., investigated changes of inflammatory mediators in CD40+ peripheral blood mononuclear cells (PMBC) in children with Autism Spectrum Disorder (ASD). The authors identified that increase of CD40+ cells that express CCR1 and multiple inflammation responding transcription factors in children with ASD. The authors utilized PBMC from ASD patients and approached their hypothesis using flow cytometry. Their findings are interesting and provide a potential clinical application as diagnostic markers and therapeutic targets for ASD. However, the depth of the data is somewhat thin and a substantial amount of supporting data is needed for publication in IJMS.
Major Points:
- The biggest concern about the manuscript is the lack of data. The authors need to provide more information about how ASD patient immunoprofiles are different from the control group. They also need to provide a logical explanation for how they specifically chose CCR1 expression. The authors indicated elevated CCR1-expressing CD40+ cells by flow cytometry. Is CCR1 the only chemokine receptor that changed in ASD? The authors need to verify the specificity. They should test other chemokines and chemokine receptors (at least some that are involved in ASD and neuroinflammation) because many of them play essential roles in inflammation induced pathologies and neuroinflammation. Furthermore, many CC class chemokines (e.g. CCL3,4,5,7) are associated with inflammation and ASD. The authors should also include these factors.
- The authors proposed that other immune responding transcription factors and cytokines are upregulated based on their flow cytometric experiments. They showed IFN-g, T-bet, IL-17A, and RORgt expressing CD40+ cells are increased in ASD patients. For a more complete test, the authors should include more immunophenotyping with more antibodies. Alternatively, single cell sequencing will provide a more complete profile.
- The authors should also test the similar phenotypes in mouse ASD model to support their findings.
Minor Points:
- The authors used simple bar graphs for quantification. I highly recommend using scatter plots for publication in a quality journal such as IJMS.
- A cartoon (model figure) showing how the CCR1 immune pathway is associated with ASD (or vice versa) will help readers understand the story in the discussion session.
- Fig 1c Western blot – Is it 3 patient samples for each group or a pool of groups? If the sample number is only 3, is the quantification and statistics reliable? They may need a larger sample size for the Western blot. Please clarify it.
- For RT-qPCR (Fig 1b), mention that it was normalized by GAPDH in the figure legend.
Reviewer 2 Report
Aldossar et al. present a study demonstrating increase of inflammatory mediators (e.g., CCR1, IFNg/T-bet, IL-17/RORg/IL-22, TNFa) on PBMC CD40+ cells. However, since the study focused only on CD40+ cells, gating on a single marker or receptor cannot fully capture the heterogeneity and complexity of immune cell populations. Furthermore, most of the gated populations are below 5% and marginally significant. The demographics and gating strategies were not provided. The data points were not transparent in box plot and no table can be traced back. Due to the inherent issue with the gating strategy which completely relies on CD40 to gate on a mixture of cell populations (B, DC, Mac, subset of T cells), the conclusion is hardly convincing with the current dataset. Thus, I cannot agree for its publication.
Reviewer 3 Report
The article ‘Upregulation of inflammatory mediators on peripheral blood CD40+ cells in children with autism spectrum disorder' by Aldossari et al. investigates the involvement of CCR1, inflammatory mediators, and transcription factor expression in CD40+ cells in ASD. The results presented here are informative and relevant for the 'IJMS.' However, the data presented here are premature for publication at this stage. The authors need to address some critical concerns in the current version of the manuscript before its publication. The issues are listed below.
The Abstract is written in very casual language. It needs to be rewritten.
The rationale behind the link between CD40 and ASD is underdeveloped. Please elaborate in the introduction.
Several studies have reported that IL-1β, , IL-6, IL-8, IL-10, IL12p40, IL-13, monocyte chemoattractant protein (MCP), IL- 23, transforming growth factor (TGF)-β1, and granulocyte macrophage colony-stimulating factor (GM-CSF)-1 as causative inflammatory mediators in ASD? Have the authors looked at these markers to understand a full spectrum of inflammatory mediators in ASD?
Can the authors comment on the correlation of individual or combinatorial cytokine profiles/inflammatory mediators with the clinical/behavioral outcomes in the patients?
While CD40 is an important marker for neuroinflammation in children with ASD, CD4+ CD25 high Tregs and CD23+ B lymphocytes are also implicated in the severe form of ASD. Have the authors looked at the inflammatory profiles of Tregs or CD23 on B lymphocytes in peripheral blood samples?
While the authors have done the flow analysis, a secondary validation of IL-22, TNF-α and other inflammatory mediators should be included.
The statistical power in all the experiments are pretty weak and invite argument that such mild differences cannot fully represent the ASD outcome.
Round 2
Reviewer 1 Report
Although no new experiments have been conducted to address the comments and produce a more data-rich article, the authors have made revisions to the data figures and provided additional explanations as suggested. These changes have improved the readability and clarity of the manuscript. I do not have any further recommendations.
Reviewer 2 Report
After careful consideration of the manuscript and the provided references, I remain unconvinced by the conclusion of this study. Given the quality and validity concerns I have identified, I would suggest that the author consider submitting their work to a more suitable journal.
Reviewer 3 Report
The reviewers have addressed most of the issues raised in the previous version of the manuscript and the unresolved issues are understandable. Hence, I endorse publication of the revised version of the paper.
